# Optimizing Multiple Entropy Thresholding by the Chaotic Combination Strategy Sparrow Search Algorithm for Aggregate Image Segmentation

**DOI:** 10.3390/e24121788

**Published:** 2022-12-06

**Authors:** Mengfei Wang, Weixing Wang, Limin Li, Zhen Zhou

**Affiliations:** 1School of Information, Chang’an University, Xi’an 710064, China; 2School of Electrical and Electronic Engineering, Wenzhou University, Wenzhou 325035, China; 3School of Optoelectronic Science and Engineering, University of Electronic Science and Technology of China, Chengdu 611731, China

**Keywords:** aggregate image, multiple entropy thresholding, sparrow search algorithm, chaotic map

## Abstract

Aggregate measurement and analysis are critical for civil engineering. Multiple entropy thresholding (MET) is inefficient, and the accuracy of related optimization strategies is unsatisfactory, which results in the segmented aggregate images lacking many surface roughness and aggregate edge features. Thus, this research proposes an autonomous segmentation model (i.e., PERSSA-MET) that optimizes MET based on the chaotic combination strategy sparrow search algorithm (SSA). First, aiming at the characteristics of the many extreme values of an aggregate image, a novel expansion parameter and range-control elite mutation strategies were studied and combined with piecewise mapping, named PERSSA, to improve the SSA’s accuracy. This was compared with seven optimization algorithms using benchmark function experiments and a Wilcoxon rank-sum test, and the PERSSA’s superiority was proved with the tests. Then, PERSSA was utilized to swiftly determine MET thresholds, and the METs were the Renyi entropy, symmetric cross entropy, and Kapur entropy. In the segmentation experiments of the aggregate images, it was proven that PERSSA-MET effectively segmented more details. Compared with SSA-MET, it achieved 28.90%, 12.55%, and 6.00% improvements in the peak signal-to-noise ratio (PSNR), the structural similarity (SSIM), and the feature similarity (FSIM). Finally, a new parameter, overall merit weight proportion (OMWP), is suggested to calculate this segmentation method’s superiority over all other algorithms. The results show that PERSSA-Renyi entropy outperforms well, and it can effectively keep the aggregate surface texture features and attain a balance between accuracy and speed.

## 1. Introduction

Aggregate particles are widely applied in civil engineering such as road traffic, railway, and housing construction. Natural aggregates, such as rough gravels and smooth pebbles, are both irregular in shape. The geometric characteristics of aggregates, such as the size, shape, and roughness, are related to the aggregate quality evaluation [1,2], and it is crucial to detect them accurately and effectively. Image processing techniques are useful for aiding aggregate detection. Since the aggregate particles are mostly obtained from a muck pile, they are often touching and overlap each other, and their surfaces are very rough. Hence, the aggregate images are usually very noisy, and the image processing for aggregates is harder than for other particles or grains.

The multi-class segmentation algorithm is a popular image processing algorithm, and it can automatically divide the multi-class homogeneous regions based on the features such as discontinuity, similarity in color or gray scale, and texture [3]. For aggregate image segmentation, thresholding, region growing, clustering, and semantic segmentation are the most commonly utilized algorithms. Among them, the watershed [4], region growing [5], and clustering algorithms [6] are typically effective for segmenting aggregate images with clear edges, but the gray-scale value transition from the aggregate surfaces to the edges is gentle. When the aggregate particles overlap or touch each other, these algorithms suffer from a severe under-segmentation problem, resulting in large particle sizes and gray-scale fusion, which loses surface roughness features. In recent years, semantic segmentation [7] has achieved impressive success in aggregate particle size detection, but the parallel segmentation for images with surface textures is still challenging. Thresholding is a straightforward algorithm, and the histogram peaks or valleys in an image are the basic features. The peaks are the most informative features in this gray-scale value range in the image, and the valleys are the least informative features. The number of peaks and troughs in a general image is not fixed, and the valleys can be used as the optimal segmentation thresholds [8]. Thresholding segmentation is simple and robust. It divides pixels into a limited number of classes based on intensity values and a set of thresholds, and it is suitable for a variety of noisy aggregate images.

Dual thresholds [9] can be employed to separate rough or smooth edges and surface textures in aggregate images. Furthermore, compared to global thresholding, multiple thresholding (MT) can easily separate the touching aggregates while maintaining the particle edges, surface roughness, and other details [10]. MT is unaffected by pixels in the neighborhood with similar gray scales. Since the gray scales of aggregate images mostly change gently, MT has better segmentation accuracy and robustness. The algorithm’s success is dependent on determining the proper thresholds. Adaptive MT, such as Otsu [11] and multiple entropy thresholding (MET) [12,13], can automatically determine thresholds. MET measures the amount of image information using entropy as the standard and solves multiple thresholds that can cause the amount of information to reach the extreme value to divide the histogram, resulting in image segmentation. Common entropy includes Renyi entropy [14], symmetric cross entropy [15], Kapur entropy [16], Tsallis entropy, exponential entropy, etc. Kapur entropy is also called maximum entropy. Renyi entropy and Tsallis entropy are extensions of Shannon entropy. They are widely utilized in object recognition and image segmentation. However, they all have the same issue that the operation time grows exponentially as the number of thresholds increases.

An optimization algorithm can assist the above MET to swiftly determine the thresholds and lower the time significantly [17,18,19]. The existing optimization algorithms include particle swarm optimization (PSO) [20], cuckoo search (CS) [21], the bat Algorithm (BAT) [22], gray wolf optimization (GWO) [23], the whale optimization algorithm (WOA) [24], the mayfly algorithm (MA) [25], the sparrow search algorithm (SSA) [26], etc. Their accuracy, speed, and stability are all affected by the population distribution and search paths [19]. For example, the search path of WOA is a spiral, causing the whales to move quickly and WOA to be fast. To prevent the sparrows from repeatedly searching the same position, the population locations of SSA are saved in a matrix. SSA divides the population into producers, scroungers, and vigilantes. These three types of sparrows simultaneously search for the optimal solution, which is quite efficient, and each sparrow has two update mechanisms, making SSA quite robust.

However, these optimization algorithms have two drawbacks, which are an incomplete global search and being trapped in local areas, resulting in a reduction in the algorithms’ accuracy. Laskar et al. [27] overcame the local stagnation by combining WOA and PSO and achieved a breakthrough in the accuracy, but the computational complexity was massively increased. Kumar et al. [28] adopted a chaotic map to make the global search more comprehensive. Later, Chen et al. [29] offered Levy flight to jump out of the local area on its basis. Although the accuracy was improved, it consumed a lot of time. At present, the algorithm with the best balance of accuracy, speed, and stability is SSA [26], which has proven to be a good parameter selector [30,31,32] in the application of network configuration [33], route planning [34], and micro-grid clustering [35]. Thus, this paper optimizes MET based on SSA and corrects the two flaws that were mentioned.

In this study, three strategies are proposed for enhancing the accuracy and stability of SSA. First, piecewise mapping was employed to make the sparrow distribution more uniform. Second, an expansion parameter was studied to increase the search range, which improves the accuracy without adding to the iteration time. Third, a range-control elite mutation strategy was put forward to ensure that the stationary sparrow leaps out of the local area, with a chance to find better values before SSA. The new algorithm is called the PERSSA. PERSSA was utilized to optimize three METs applied to aggregate images. They were Renyi entropy, symmetric cross entropy, and Kapur entropy. This model is called PERSSA-MET.

In short, the main contributions of this paper are listed as follows:

1. The PERSSA-MET is proposed for the segmentation of aggregate images, which can effectively preserve the surface roughness and edge features of aggregates and achieve the best balance between accuracy and speed.

2. Aiming at the characteristic that the aggregate image histogram changes smoothly but has many extreme points, a novel expansion parameter and range-control elite mutation strategies were studied, which can effectively jump out of the local optimum. Combining them with piecewise mapping improved the optimization accuracy and stability of SSA. This algorithm is called PERSSA.

3. A comparative experiment was carried out on three popular METs, which were Renyi entropy, symmetric cross entropy, and Kapur entropy, and the results show that PERSSA-Renyi Entropy is better for aggregate image segmentation.

4. To comprehensively evaluate all methods, an overall merit weight proportion (OMWP) was created to quantify the dominance of the algorithm in all algorithms, which combined precision, stability, and speed.

The remainder of this paper is structured as follows: Section 2 explains the basics of MET and SSA. Section 3 describes PERSSA-MET in detail, including PERSSA and its process for optimizing MET. Section 4 verifies the performance and effectiveness of PERSSA and PERSSA-MET through benchmark function tests and segmentation experiments on various aggregate images. Finally, Section 5 concludes the study and looks forward to future work.

## 2. Related Works

In this section, the threshold determination methods of three METs are first introduced, and SSA and its current development are described.

### 2.1. MET

Multiple entropy thresholding (MET) utilizes a histogram to classify pixels into categories based on gray scale and assigns the nearest gray scale to each category. Common METs are Renyi entropy [14], symmetric cross entropy [15], and Kapur entropy [16]. Each MET determines the information differently.

When the number of thresholds is K, the histogram is divided into K+1 regions. The information of these regions is $H_{1}$, $H_{2}$, ⋯, $H_{K+1}$, and the total information is $E=H_{1}+H_{2}+\cdots +H_{K+1}$.

The information amounts for Renyi entropy, symmetric cross entropy, and Kapur entropy in the *k*-th region can be expressed as Equations (1)–(3):(1)$$H_{K-Renyi}\left(l\right)=\frac{1}{1-\alpha }ln\left(\sum_{i=l_{K-1}}^{l_{K}}{\left(\frac{P_{i}}{\omega_{K}\left(l\right)}\right)}^{\alpha }\right)\;$$
(2)$$H_{K-Symmetric\;Cross}\left(l\right)=\sum_{i=l_{K-1}}^{l_{K}}h_{i}(i\cdot ln\left(\frac{i}{u_{K}\left(l\right)}\right)+u_{K}\left(l\right)\cdot ln\left(\frac{u_{K}\left(l\right)}{i}\right))\;$$
(3)$$H_{K-Kapur}\left(l\right)=-\sum_{i=l_{K-1}}^{l_{K}}\frac{P_{i}}{\omega_{K}\left(l\right)}ln\left(\frac{P_{i}}{\omega_{K}\left(l\right)}\right)\;$$
where $\left[l_{K-1},\;l_{K}\right]$ is the gray-scale range of the K-th region, $0\leq l_{K-1}\leq l_{K}\leq L$, L is the gray scale of the image, $h_{i}$ is the frequency with grey-scale is i, $u_{K}$ is the mean value of the gray-scale of the region, $P_{i}$ is the probability that the gray scale is i, $P_{i}\in \left[0,1\right],$ $\omega_{K}$ is the sum of the probabilities within the region, $\omega_{K}=\overset{l_{K}}{\underset{i=l_{K-1}}{\sum }}P_{i}$, α∈[0, 1) is a tunable parameter, and we take α=0.5.

To calculate the $l_{best}\left(1,\;2,\;\cdots ,\;K\right)$ that makes E reach its extremum values, which are the final segmentation thresholds, the functions corresponding to the three METs are presented in Equations (4)–(6):(4)$$l_{best-Renyi}\left(1,2,\ldots ,k\right)=argmax\left\{E\right\}$$
(5)$$l_{best-Symmetric\;Cross}\left(1,2,\ldots ,k\right)=argmin(E\}$$
(6)$$l_{best-Kapur}\left(1,2,\ldots ,k\right)=argmax\left\{E\right\}$$

These three METs have varied segmentation effects on the objects or histograms of distinct features [3,36], and Table 1 illustrates their differences in aggregate image segmentation. Since there are too many particles for an intuitive review, only a part of the aggregate image, its histogram, and its segmentation results at K=2 are shown.

Renyi entropy and Kapur entropy detected more bright details for black aggregates, whereas the symmetric cross entropy recognized more shadow details, but its emphasis on detecting white aggregates was the inverse. The main reason for this difference is the changes in the histogram caused by the color, texture, and target ratio of the aggregate image.

The histogram of the aggregate image normally fluctuates smoothly at the peaks and valleys, but there are several extreme points, which are the aggregate characteristics. They are close, but if the thresholds differ only by one gray-scale value, the segmentation results may be quite dissimilar. Therefore, the accuracy and stability of the segmented images are directly influenced by the performance of the optimization algorithm.

### 2.2. SSA

The sparrow search algorithm (SSA) [26] is an optimization algorithm that mimics sparrow foraging and anti-predation behavior. The sparrow position $x_{i,j}$ affects its fitness value $f\left(x_{i,j}\right)$, where i∈[1, n] is the i-th in n sparrows and j∈[1, d] is the j-th in the d-dimensional search space.

During foraging, the sparrows are separated into producers and scroungers. The producers have better fitness values, and they provide the foraging area and direction. The producers update their positions according to Equation (7):(7)$$x_{i,j}^{t+1}=\{\begin{array}{ll} x_{i,j}^{t}\cdot exp\left(-\frac{i}{\alpha \cdot T_{max}}\right) & R_{2}<ST\\ x_{i,j}^{t}+Q\cdot L & R_{2}\geq ST \end{array}$$
where t is the current number of iterations, $T_{max}$ is the maximum number of iterations, α∈(0, 1] is a random number, $R_{2}\in \left(0,\;1\right]$ is the alarm value, ST∈[0.5, 1] is the safe value, Q is a random number that obeys a normal distribution, and L is the all-ones matrix of order 1×d.

The scroungers follow the producers to obtain food, but some extremely hungry scroungers will convert their foraging paths. The scroungers update their positions according to Equation (8):(8)$$x_{i,j}^{t+1}=\{\begin{array}{ll} x_{best}^{t+1}+\left|x_{i,j}^{t}-x_{best}^{t+1}\right|\cdot A^{+}\cdot L & i\leq n/2\\ Q\cdot exp\left(\frac{x_{worst}^{t}-x_{i,j}^{t}}{i^{2}}\right) & i>n/2 \end{array}$$
where $x_{best}$ is the optimal position, $x_{worst}$ is the worst position, A is the matrix of order 1×d, its element value is ±1, and $A^{+}=A^{T}{\left(AA^{T}\right)}^{-1}$.

During anti-predation, the vigilantes are randomly generated. The sparrows on the edge quickly move to the safe location, and the remaining sparrows approach each other. The vigilantes update their positions according to Equation (9):(9)$$x_{i,j}^{t+1}=\{\begin{array}{ll} x_{best}^{t}+\beta \cdot \left|x_{i,j}^{t}-x_{best}^{t}\right| & f_{i}>f_{best}\\ x_{i,j}^{t}+K\cdot \frac{\left|x_{i,j}^{t}-x_{worst}^{t}\right|}{\left(f_{i}-f_{\omega orst}\right)+\varepsilon } & f_{i}=f_{best} \end{array}$$
where β is the step size control parameter, which obeys the standard normal distribution; K∈[−1, 1] controls the sparrows’ movement direction; ε is the smallest constant; $f_{best}$ is the best fitness value; and $f_{\omega orst}$ is the worst fitness value.

Figure 1 shows the optimization principle of SSA. The optimization accuracy and speed of SSA are directly related to the sparrows’ position distribution, search path, and local optimal solution. Some scholars have proposed evolutionary strategies for these three points. Lv et al. [37] introduced the bird swarm algorithm in the SSA’s producers, and the precision increased, but the speed decreased. Chen et al. [29] combined a chaotic map, dynamic weight, and Levy flight with SSA (CDLSSA) to achieve better optimization accuracy and stability, but the segmentation time was lengthened. To ensure speed, Liu et al. [34] proposed to utilize a chaotic map and the adaptive inertia weight optimization SSA (CASSA), but the accuracy improvement was minor. Currently, there is no strategy that can achieve the best balance of segmentation accuracy and convergence speed for the SSA.

## 3. Proposed Method

In this section, the proposed PERSSA is first introduced, and the specific steps of PERSSA to optimize MET for segmented aggregate images are described.

### 3.1. PERSSA

PERSSA combines piecewise mapping with the expansion parameter and range-control elite mutation proposed in this paper for the first time, which can effectively jump out of the local optimum and improve the accuracy and stability of the SSA without reducing the convergence speed.

#### 3.1.1. Piecewise Mapping

In the SSA, the sparrows’ initial positions are random, and clustered sparrows are terrible for a global search. Therefore, it is recommended to introduce a chaotic map in the population initialization process to increase the randomness and uniformity of the initial position. Varol et al. [38] compared ten chaotic maps, such as the circle, logistic, piecewise, singer, and tent mapping. Piecewise mapping is the most accurate and stable among them, and it can quickly disturb the population without increasing the optimization time. It can be used to perturb the sparrows’ initial positions. Piecewise mapping can be described by Equation (10):(10)$$x\left(k+1\right)=\{\begin{array}{cc} \frac{x\left(k\right)}{P} & 0\leq x<P\\ \frac{x\left(k\right)-P}{0.5-P} & P\leq x<0.5\\ \frac{1-P-x\left(k\right)}{0.5-P} & 0.5\leq x<1-P\\ \frac{1-x\left(k\right)}{P} & 1-P\leq x<1 \end{array}$$
where P∈(0, 1) is the control parameter and P≠0.5. Its value affects the randomness and uniformity of the sequence. Figure 2 is the chaotic sequence of P=0.4, 0.6, 0.8 when x(1)=0.1.

It can be seen that the Piecewise mapping has the strong randomness. The closer the P is to 1, the more non-uniform the sequence. The chaotic sequence is most uniform when P=0.4.

The steps for piecewise mapping to perturb the sparrows’ initial positions are as follows:

(1) Generate values: $x_{i,j}\left(0\right),\;\left(i=1,\;2,\;\cdots ,\;n;\;j=\left(1,\;2,\;\cdots ,\;d\right)\right)$ randomly in [0, 1].

(2) Set the P value and generate the chaotic sequence through Equation (10). We take P=0.4.

(3) Convert the $x_{i,j}$ range from [0, 1] to [lb, ub]. $x_{i,j}^{\prime }=lb+x_{i,j}\cdot \left(ub-lb\right)$, where ub and lb are the upper bound and lower bound of the search space, respectively. Take $x_{i,j}^{\prime }$ as the sparrows’ initial positions.

#### 3.1.2. Expansion Parameter

The producers are affected by the function $y=e^{-x}$. As the number of iterations increases, the search scope narrows rapidly. This reduces the global search capability, and it is easy to fall into a local region, reducing the optimization precision. Thus, an expansion parameter, σ, is proposed to widen the search range. At an early stage of the iteration $t<T_{max}/2$, σ is larger and spreads the sparrows as far as possible. At a late stage of the iteration $t\geq T_{max}/2$, the sparrows are concentrated. Therefore, σ is smaller. The expansion parameter can be expressed by Equation (11):(11)$$\sigma =\{\begin{array}{cc} \sigma_{max}-\frac{1+cos\left(t\pi /T_{max}\right)}{2} & t<T_{max}/2\\ \sigma_{min}+\frac{1-cos\left(t\pi /T_{max}\right)}{2} & t\geq T_{max}/2 \end{array}$$
where $\sigma_{max}$ is the parameter at the beginning of the iteration and $\sigma_{min}$ is the parameter at the end of the iteration. After experiments, the expansion effect is the best when $\sigma_{max}$ is close to 1 and $\sigma_{min}$ is close to −1. We take $\sigma_{max}=0.99$ and $\sigma_{min}=-0.99$.

By adding Equation (11) into the producers’ location update, Equation (7) becomes Equation (12).
(12)$$x_{i,j}^{t+1}=\{\begin{array}{ll} x_{i,j}^{t}\cdot exp\left(-\frac{i}{\sigma \cdot \alpha \cdot T_{max}}\right) & R_{2}<ST\\ x_{i,j}^{t}+Q\cdot L & R_{2}\geq ST \end{array}$$

#### 3.1.3. Range-Control Elite Mutation

The sparrows fall into many locally optimal solutions in the search. If the individual cannot escape in time, the population is excessively consumed, resulting in better solutions being missed. The existing strategies, such as Cauchy–Gaussian mutation, Levy flight, and random walk, do not ensure that the sparrow leaves the neighborhood, and some of them take more time.

Hence, the range-control elite mutation strategy is studied, which can be implemented quickly while improving the precision of the solution. Each iteration selects an elite sparrow ($x_{best}$) with the highest fitness value ($f_{best}$) to jump out of the local region and regulates the distance it moves. The sparrows are still scattered when $x_{best}$ is far from $x_{worst}$ with the lowest fitness value so that $x_{best}$ is randomly mutated within (lb, ub). Conversely, the sparrows are assembled when $x_{best}$ and $x_{worst}$ are closer together, controlling $x_{best}$ to mutate outside the rectangular area of $x_{best}-x_{worst}$ and inside the region of (lb, ub). This is performed to ensure that the elite sparrow ($x_{best}$) continues to optimize globally while avoiding the local optimum. The elite sparrow updates its position according to Equation (13).
(13)$$x_{best}^{\prime }=\{\begin{array}{ll} \left(ub-lb\right)\cdot randn+lb & \left|x_{best}-x_{worst}\right|>\frac{ub-lb}{2}\\ 2x_{best}-x_{worst}+\left(\left(ub-lb\right)\cdot randn+lb\right)\cdot randn & \left|x_{best}-x_{worst}\right|\leq \frac{ub-lb}{2} \end{array}$$

This strategy can greatly increase the probability of the sparrow jumping out of the local optimal solution, and it does not consume iteration time compared with the traditional strategies.

### 3.2. PERSSA-MET

PERSSA can effectively address the issues of low MET exhaustive method efficiency and the low accuracy and stability of related improvement strategies. In the image segmentation, PERSSA employs MET as the objective function and the image histogram as the search space. PERSSA distributes sparrows globally and converges quickly to allow the function to achieve an extreme value, and the matching sparrow positions are the segmentation thresholds. This model is called PERSSA-MET. The related processes for segmenting aggregate images with PERSSA-MET are illustrated in Figure 3.

The red words in Figure 3 represent the innovation points of the research. The detailed PERSSA process is shown in the dashed box, which is divided into three stages: (1) parameter initialization, including the parameters of PERSSA, parameters extended from MET, and the population’s initial position, x(0), after the chaotic mapping; (2) position update: the sparrows are divided based on the fitness values, and three equations are applied to update the positions of the three kinds sparrows; and (3) iterative and mutation: the greedy algorithm keeps the better solutions at each update, and each iteration selects the global optimal sparrow to perform the range-control elite mutation. Until the last iteration, the population converges, the final output $f\left(x_{best,\;d}\right)$ is employed to assess the benefits and the drawbacks of the optimization algorithm, and $x_{best,\;d}$ is used as the threshold to segment the aggregate image.

In Figure 3, PD is the proportion of producers and SD is the proportion of vigilantes.

## 4. Experiments and Analyses

In this section, on the one hand, PERSSA and the other seven existing similar optimization algorithms were evaluated through benchmark function tests. On the other hand, experiments on aggregate image segmentation assess were conducted on the performance of PERSSA-MET.

For a fair comparison, all experiments were performed on a PC equipped with an Intel (R) Core (TM) i5-10400F @2.90 GHz CPU and 16 GB RAM, and they were implemented using MATLAB-R2018b within Win-10.

### 4.1. Evaluation of PERSSA

The benchmark function test was utilized to evaluate the performance of the optimization algorithms, and six benchmark functions were selected, as shown in Table 2. The uni-modal function has only one optimal value, which can test the local convergence ability of the optimization algorithm, and the multi-modal function has several local optimal values and one global optimal value, which can assess the global search ability of the optimization algorithm. The dimension (*D*) of the benchmark functions was uniformly set to 30, and the versions are illustrated in Figure 4 when the dimension is 2.

PERSSA was compared with seven algorithms, which were PSO [20], GWO [23], WOA [24], MA [25], SSA [26], CDLSSA [29], and CASSA [34]. To ensure fairness, some parameters were set to maximize the performance of each optimization algorithm. They were a population size of 60 and a maximum iteration number of 600. Furthermore, in PSO, $C_{1}=C_{2}=1.5$ and ω=0.74; in GWO, α decreased linearly from 2 to 0 and $r_{1},\;r_{2}\in \left[0,\;1\right]$; in WOA, α∈[0, 2], b=1, and l∈[−1, 1]; in MA, g=1, gdamp=1, $\alpha_{1}=1$, and $\alpha_{2}=\alpha_{3}=1.5$; in SSA, CDLSSA, CASSA, and PERSSA, PD=20%, SD=10%, and ST=0.8; in CDLSSA levybeat=1.5 and K=2; and in CASSA S=1.

The best value (*Best*), the average value (*Avg*), the standard deviation (*SD*), and the time consumption (*T*) were selected as the evaluation indicators based on the accuracy and speed of the optimization algorithm, and the unit of *T* is seconds. Since the sparrows’ initial positions are random, the average value of 60 optimization experiments was taken for the *Best*, *Avg,* and *SD*, and *T* is the total time spent during the experiment. The relevant statistics are provided in Table 3, in which the bold characters are the best values when comparing the seven optimization algorithms horizontally.

It can be seen from Table 3 that PERSSA had a better overall performance. On the uni-modal functions $F_{1}$~$F_{3}$, the *Best*, *Avg,* and *SD* of PERSSA were significantly higher than those of the other algorithms, and its *T* was only slightly lower than the optimal *T*. On the multi-modal functions $F_{4}$~$F_{6}$, PERSSA was still superior, and its *T* was optimal on $F_{6}$. In this experiment, WOA, SSA, CASSA, and PERSSA were faster, while CDLSSA and PERSSA had the highest accuracy and stability. However, the *T* of CDLSSA was roughly twice that of PERSSA. Taken together, PERSSA achieved the best balance of speed and precision.

Figure 5 displays the convergence curves of the eight algorithms in a random experiment to observe the convergence process of PERSSA. For the 30-dimensional functions, PERSSA had the highest accuracy with fewer iterations and converged in approximately 200 iterations. The accuracy of CDLSSA and PERSSA were the same on $F_{1}$, $F_{2}$, and $F_{5}$, but one iteration time of CDLSSA was too long and was inefficient. On $F_{3}$, $F_{4}$, and $F_{6}$, the initial values of PERSSA were closer to the optimal value, which was attributed to the addition of piecewise mapping during the population initialization to make the sparrows’ positions more uniform. It can be observed from $F_{1}$ and $F_{2}$ that PERSSA had a wider search scope. After adding the expansion parameter, the local optimal value was continuously updated and was close to the global optimal value. On all benchmark functions, the convergence curves of PERSSA had polylines. This was due to the inclusion of the range-control elite mutation, which can easily jump out of local areas, even in the $F_{4}$ and $F_{6}$ dilemmas. Although CASSA had similar polylines, the escape effect was poor. Overall, when the three proposed strategies work together, PERSSA not only has high precision and fast speed but can also be applied to various functions with high robustness.

The Wilcoxon rank-sum test [39] can compare the correlation between two samples. If P-value≤0.05, there is a significant difference. Since the sample size in Table 3 was small and the dimension was single, the parameters of 2 and 60 dimensions were added in the inspection to make the experimental results more accurate. The Wilcoxon rank-sum test results of PERSSA and the other optimization algorithms are shown in Table 4, where the bold characters represent significant differences between the two samples.

The results show that the conclusions were consistent with the previous ones. PERSSA’s precision, stability, and robustness differed significantly from those of PSO, GWO, WOA, and MA. PERSSA’s stability differed significantly from that of SSA and CASSA, and its speed differed significantly from that of CDLSSA. From SSA to CASSA to CDLSSA, the precision and robustness increased. PERSSA had the smallest difference in precision compared to the most accurate algorithm (CDLSSA), and it was faster than WOA, SSA, and CASSA. To summarize the result: PERSSA effectively improved the precision while attaining the optimal balance between accuracy and speed.

### 4.2. Performance of PERSSA-MET

Three METs were combined with PERSSA and SSA to realize PERSSA-MET (i.e., PERSSA-Renyi entropy, PERSSA-symmetric cross entropy, and PERSSA-Kapur entropy) and SSA-MET (i.e,. SSA-Renyi entropy, SSA-symmetric cross entropy, and SSA-Kapur entropy). In Section 4.1, PERSSA was iterated 200 times to achieve convergence on the 30-dimensional functions. After several experiments, only 100 iterations could maximize the savings in the segmentation time while ensuring the accuracy, and the other parameter settings were the same as those in Section 4.1.

The images utilized in the experiment were all obtained from the Key Laboratory of Road Construction Technology and Equipment, Ministry of Education, and they were RGB images with 512 × 512 pixels. A total of 100 aggregate images were tested. Table 5 shows five images with different features, and Table 6 shows their histograms. These histograms are usually flat at the peaks or valleys. For example, No. 1 and No. 4 show gentle gray-scale changes in the dark region without obvious valleys; No. 2 and No. 5 show gentle gray-scale changes in the bright area without evident peaks; and No. 3 has multiple peaks and troughs, and their distances are very short. For rough aggregate surfaces, the segmentation results are mostly discrete granular. Even if the segmentation thresholds are close, many points may be blurred to reduce the original roughness. Thus, the performance of the optimization algorithm directly affects the segmentation accuracy.

Aggregate images contain abundant information. These aggregate particles have different sizes, shapes, colors, surface roughness, etc., and these characteristics are applied to judge the coarse and fine aggregates or gravels and pebbles, to analyze the mineral types, and to decide the grinding degree, etc. Generally, the categories to be segmented are determined according to the application scenarios. The number of thresholds (K) is set, and the image is divided into K+1 categories, which is also the number of fuzzy C-means (FCM) clustering centers. In most of the related literature, K=2~6, and this K can evaluate the accuracy of thresholds while meeting the basic engineering requirements. Due to the presence of space limitations, this paper only presents the partial segmentation results of K={2, 4, 6}, as shown in Table 5. The colored boxes in Table 5 show areas with large differences in the segmentation of the aggregate surface roughness features. The blue boxes represent better segmentation results, and the red boxes represent the segmentation results that were seriously distorted.

Subjectively, FCM performed the best segmentation when K=2, followed by Renyi entropy. As K increased to 4, the accuracy of the MET segmentation results was improved, while FCM became more unstable, and the gray scale of the adjacent pixels merged. When K=6, Renyi entropy still performed well for the aggregate images with small particle sizes and touching or overlapping particles, followed by Kapur entropy. Meanwhile, the overall gray scale of symmetric cross entropy was higher. Although the edge of FCM was clear, the gray-scale deviation was too large, and the surface roughness features were lost.

Table 6 shows the histograms of the segmentation results of PERSSA-Renyi entropy, and those of FCM are shown in Table 5. The closer the distribution of the histogram of the segmented image to the histogram of the original image, the better. When K=2, the segmentation result of FCM was closer to the original image, but when K=4, 6, the segmentation results of MET were closer to the original image. For the aggregate image, when the number of thresholds was large, the similarity between the segmentation result and the original image was high. At this time, the histogram can be used as one of the methods to detect the segmentation result.

Table 7 shows the thresholds and fitness values calculated by segmenting the same image when K=6 for PERSSA-MET and SSA-MET. The thresholds are the $l_{best}\left(1,\;2,\;\cdots ,\;K\right)$ in Section 2.1, which were utilized to segment the images. This parameter cannot evaluate the pros and cons of the algorithm. It can only indicate that there are differences when the algorithm divides the image. Each set of thresholds corresponds to a unique fitness value, which is used to evaluate the algorithm, that is, the *E* value in Section 2.1. The higher fitness values of Renyi entropy and Kapur entropy are better, while the lower fitness values of symmetric cross entropy are better. The better of the two fitness values in Table 7 are highlighted in bold.

In Table 7, the ratio of PERSSA and SSA to obtain better fitness values was 15:0, which proved that PERSSA has a superior optimization ability. In the experiment, it was found that when K=2, PERSSA and SSA may obtain the same fitness value at different thresholds, but the segmentation results were slightly different. However, as K increased, the performance of PERSSA improved, especially at K=6, and PERSSA-MET outperformed SSA-MET in both fitness values and segmentation quality.

The optimization algorithm’s performance had a significant impact on the accuracy and stability of the segmentation results. Figure 6 and Figure 7, respectively, show the segmentation results of No. 2 and No. 5 on the aggregate image using seven algorithms when K=6. The white box is the typical rough texture information of the aggregate surface. It can be seen that the PERSSA-MET segmented image has more detailed features, and the color is closer to the original image, whereas the segmented images of SSA-MET lack many details, especially the aggregate surfaces. This mistake can result in erroneous roughness grading, which can waste material resources by mistakenly grinding or sandblasting.

The quality evaluation of a segmented image needs to consider the category of each pixel. The naked eye cannot label the ground truth of the aggregate image under each K. Therefore, the segmented image was compared with the original image.

The peak signal-to-noise ratio (*PSNR*), structure similarity (*SSIM*), feature similarity (*FSIM*), and time consumption (*T*) were all used as evaluation indicators. Due to the randomness of PERSSA and SSA during the position initialization, the average (*Avg*) and standard deviation (*SD*) of 60 experiments were used as the final experimental result. In Table 8, Table 9 and Table 10, the bold font represents the evaluation parameter value that is better between PERSSA-MET and SSA-MET, and the same value is not marked.

*PSNR* is a metric for measuring image noise, and its unit is dB. The larger the *PSNR* value, the more information, the less noise content, and the better the segmentation effect.
(14)$$PSNR=20{log}_{10}\left(\frac{255}{RMSE}\right)$$

The root-mean-square error (*RMSE*) is the distance between pixels:(15)$$RMSE=\sqrt{\frac{\sum \sum {\left(f\left(i,j\right)-\hat{f}\left(i,j\right)\right)}^{2}}{M\times M}}$$
where f(i,j) is the original image and $\hat{f}\left(i,j\right)$ is the segmented image.

*SSIM* was used to compare the overall similarity between the segmented image and the original image. SSIM∈[0, 1], and the higher the *SSIM* value, the smaller the distortion:(16)$$SSIM\left(I,\hat{I}\right)=\frac{\left(2\mu_{f}\mu_{\hat{f}}+C_{1}\right)\times \left(2\sigma_{f,\hat{f}}+C_{2}\right)}{\left(\mu_{f}{}^{2}+\mu_{\hat{f}}{}^{2}+C_{1}\right)\times \left(\sigma_{f}{}^{2}+\sigma_{\hat{f}}{}^{2}+C_{2}\right)}$$
where $\mu_{f}$ and $\mu_{\hat{f}}$ are the average gray-scale values of the original image and the segmented image, respectively; $\sigma_{f}$ and $\sigma_{\hat{f}}$ are the standard deviations of the original image and segmented image, respectively; $\sigma_{f,\hat{f}}$ is the covariance; and $\;C_{1}$ and $C_{2}$ are the constants utilized to avoid instability at $\mu_{f}{}^{2}+\mu_{\hat{f}}{}^{2}\approx 0$. We take $C_{1}=C_{2}=$ 6.45.

*FSIM* compares the feature differences between images before and after segmentation. Phase consistency (*PC*) extracts stable features in the local structures, and gradient magnitude (*GM*) characterizes the direction. FSIM∈[0, 1], and the higher the *FSIM* value, the better quality of the segmented images:(17)$$FSIM=\frac{\sum S_{L}\times PC_{m}}{\sum PC_{m}}$$
where $S_{L}$ is the similarity score, $S_{L}\left(w\right)=S_{PC}S_{G}$, $S_{PC}=\frac{2PC_{1}PC_{2}+T_{1}}{PC_{1}^{2}+PC_{2}^{2}+T_{1}}$, $S_{G}=\frac{2G_{1}G_{2}+T_{2}}{G_{1}^{2}+G_{2}^{2}+T_{2}}$, G is the image gradient, $T_{1}$ and $T_{2}$ are the constants, and we take $T_{1}=0.85$ and $T_{2}=160$.

In Table 8, Table 9 and Table 10, the bold font represents the evaluation parameter value that is the better value between PERSSA and SSA, and the same value is not marked.

In Table 8, Table 9 and Table 10, the bold font represents the better value between PERSSA-MET and SSA-MET, and the statistical results of the number of each table are displayed in the penultimate row of the table. It can be known that PERSSA was better than SSA in the optimization of the three METs, which means that PERSSA-MET can segment more aggregate details than SSA-MET.

The blue word represents the optimal value of the horizontal comparison of the seven algorithms, and the statistical data are displayed in the last row of the table. The blue number ratio of the seven algorithms is 31:16:16:9:8:0:22. In general, PERSSA-Renyi entropy was relatively better.

In Table 8, SSA-symmetric cross entropy had more blue numbers than PERSSA-symmetric cross entropy, and their bold numbers were similar. The analysis was affected by the standard deviation (*SD*).

Figure 8 counts the *SD* of all of the algorithms. The smaller the standard deviation, the more stable the algorithm. Therefore, symmetric cross entropy had the best stability, followed by Renyi entropy. Kapur entropy was slightly less stable, and FCM was the least stable. In summary, the stability of PERSSA-MET was higher than that of SSA-MET.

The above is the evaluation of segmentation accuracy and stability. The running speed of the algorithm also needs to be considered in the application. In Table 11, the average time (*T)* for the algorithm to segment an image is counted, and the unit of *T* is seconds.

Comparing PERSSA with SSA, the better *T* value ratio was 6:3 (Table 11). The three strategies of PERSSA did not reduce the efficiency of SSA. Conversely, PERSSA was faster than SSA in optimizing symmetric cross entropy. The main reason lies in the particularity of the aggregate image histogram, which was not only gentle at the valley but also had many extreme points, leading to many locally optimal solutions in the search process. SSA is prone to fall into these local areas, whereas PERSSA can jump out of the local areas in time with the range-control elite mutation. Therefore, PERSSA-MET is more suitable for aggregate image segmentation.

Figure 9 shows a line chart of the evaluation values of the seven segmentation algorithms under four parameters. For the same segmentation method, it was not always optimal in the four parameters, which brings trouble to the practical evaluation of the algorithms.

Hence, *PSNR*, *SSIM*, *FSIM*, and *T* were fused into a new parameter, called the overall merit weight proportion (*OMWP*), which was used to represent the superiority degree of each algorithm among all algorithms, and OMWP∈[0, 1]. This parameter was the comprehensive evaluation result of the precision, stability, and speed. It met the application requirements. The *OMWP* value of algorithm *I* can be calculated by Equation (18), and the *OMWP* values of all the segmentation algorithms are summarized in Table 12.
(18)$$OMWP\left(I\right)=\frac{1}{n\left(i\right)\times n\left(j\right)}\sum \sum \frac{\left|\bar{I_{i,\;j}}-worst_{i,\;j}\right|}{best_{i,\;j}-worst_{i,\;j}}$$
where i=2, 4, 6 are the numbers of the thresholds; j=PSNR(Avg), SSIM(Avg), FSIM(Avg), PSNR(SD), SSIM(SD), FSIM(SD), and T are the evaluation parameters; the worst situation is the worst value, such as the minimum values in *PSNR(Avg)*, *SSIM(Avg),* and *FSIM(Avg)* and the maximum values in *PSNR(SD)*, *SSIM(SD)*, *FSIM(SD),* and *T*; best is the optimal value, the opposite of worst; and n(x) is the number of x. In this study, n(i)=3 and n(j)=7.

In Table 12, the OMWP values of PERSSA-MET were always higher than those of SSA-MET. Moreover, the PERSSA-Renyi entropy score was the best one, which proves that this method has the greatest practicability in the segmentation of aggregate images.

Figure 10 shows the segmentation results and segmentation threshold lines of PERSSA-Renyi entropy. It can be seen that the selection of the threshold was reasonable and the algorithm had high segmentation accuracy. In practical applications, the number of thresholds can be selected according to the requirements.

## 5. Conclusions

This paper proposed a segmentation model (i.e., PERSSA-MET) for aggregate images that effectively preserved the rough texture and edge features of the aggregate surface. It consisted of the swarm intelligence optimization algorithm PERSSA and multiple entropy thresholding (MET). First, the three evolutionary strategies of PERSSA were specially proposed for the flat and multi-extreme points of the aggregate image histogram. Piecewise mapping made the location distribution of the sparrows more uniform and random, the expansion parameter was suggested to expand the search range, and the range-control elite mutation strategy could effectively jump out of the local area, which greatly improved the optimization accuracy and stability of SSA. Then, PERSSA was utilized to swiftly calculate the MET thresholds, overcoming the disadvantage of MET’s long operation time as the number of thresholds (*K)* increases. The experiments compared the performance of three METs in aggregate image segmentation. In terms of precision, Renyi entropy was first, Kapur entropy was second, and symmetric cross entropy was third. In terms of stability, symmetric cross entropy was first, Renyi entropy was second, and Kapur entropy was third. Finally, in order to comprehensively evaluate the pros and cons of the algorithm, an evaluation parameter, OMWP, which combines segmentation precision, stability, and operation speed was studied and was comprehensive and fair. PERSSA-Renyi entropy achieved the highest OMWP values, with an optimal balance between precision, stability, and speed.

In future work, efforts will be made to enhance the image segmentation accuracy and to take these results as the input of deep learning to achieve the parallel classification of various aggregate features. Furthermore, this segmentation model, PERSSA-MET, can be extended to similar fields, such as tissue or cereal images, and has broad application prospects.

## Figures and Tables

**Figure 1 entropy-24-01788-f001:**
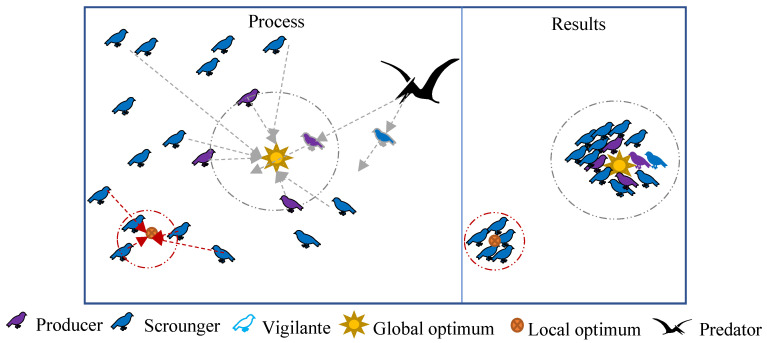
SSA optimization schematic.

**Figure 2 entropy-24-01788-f002:**
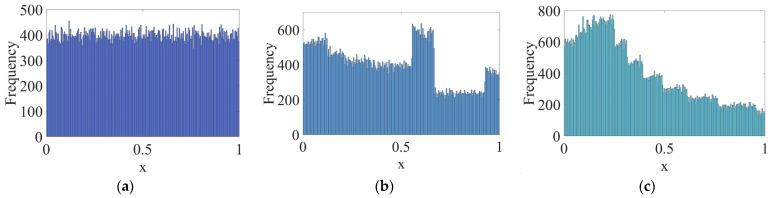
Chaotic sequences of piecewise mapping: (**a**) P=0.4; (**b**) P=0.6; (**c**) P=0.8.

**Figure 3 entropy-24-01788-f003:**
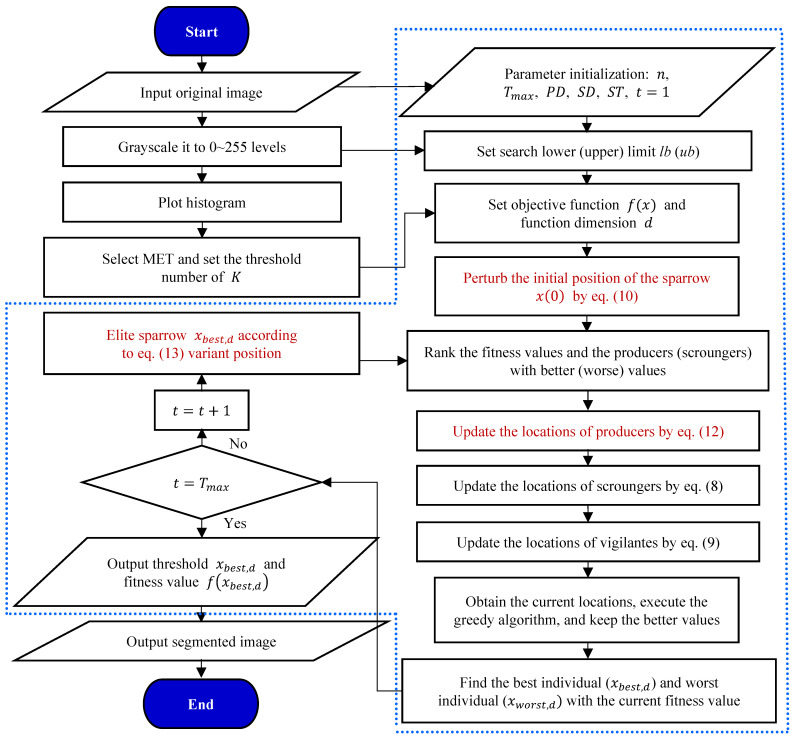
Flowchart for segmenting aggregate images by applying PERSSA-MET.

**Figure 4 entropy-24-01788-f004:**
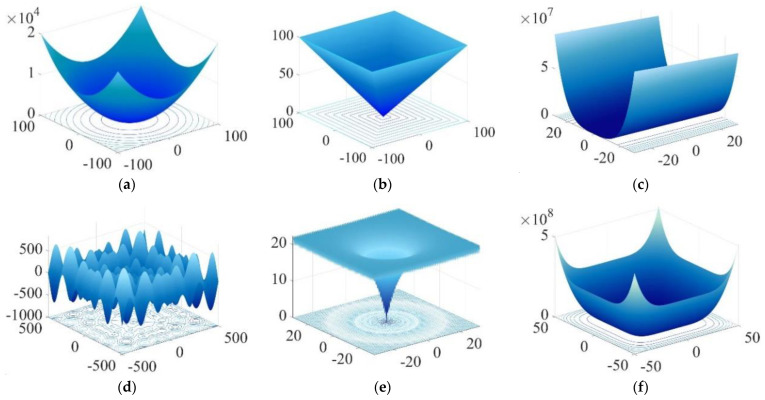
Two-dimensional versions of the six benchmark functions: (**a**–**f**) correspond to $F_{1}\left(x_{1},x_{2}\right)$–$F_{6}\left(x_{1},x_{2}\right)$.

**Figure 5 entropy-24-01788-f005:**
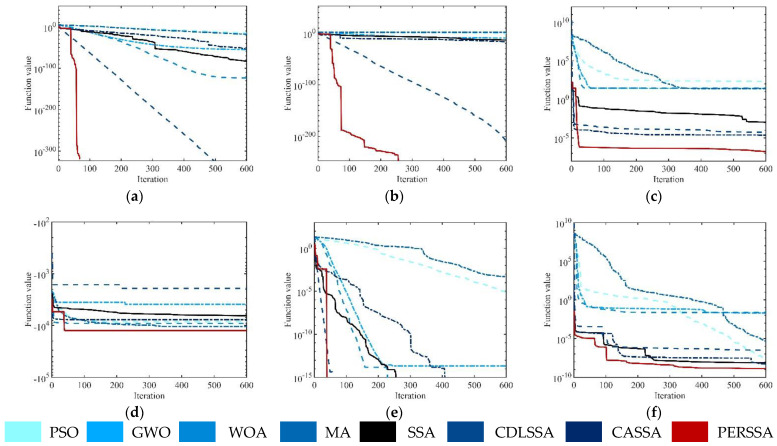
Iteration curves of the optimization algorithms on the benchmark functions: (**a**–**f**) $F_{1}$–$F_{6}$. The abscissa is the number of iterations, and the ordinate is the value of the objective function.

**Figure 6 entropy-24-01788-f006:**
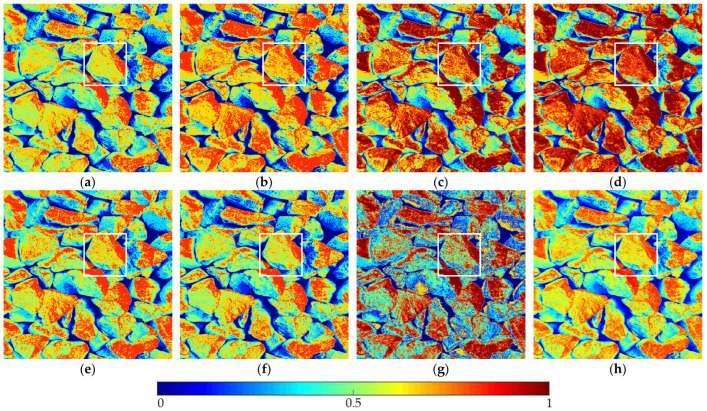
The segmentation results of No. 2 by PERSSA-MET and SSA-MET when K=6: (**a**) PERSSA-Renyi entropy; (**b**) SSA-Renyi entropy; (**c**) PERSSA-symmetric cross entropy; (**d**) SSA-symmetric cross entropy; (**e**) PERSSA-Kapur entropy; (**f**) SSA-Kapur entropy; (**g**) FCM; (**h**) original image. White boxes are areas where roughness differences are noticeable.

**Figure 7 entropy-24-01788-f007:**
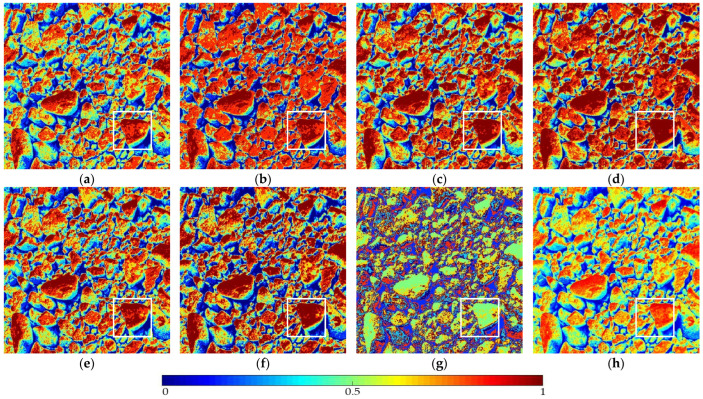
The segmentation results of No. 5 by PERSSA-MET and SSA-MET when K=6: (**a**) PERSSA-Renyi entropy; (**b**) SSA-Renyi entropy; (**c**) PERSSA-symmetric cross entropy; (**d**) SSA-symmetric cross entropy; (**e**) PERSSA-Kapur entropy; (**f**) SSA-Kapur entropy; (**g**) FCM; (**h**) original image. White boxes are areas where roughness differences are noticeable.

**Figure 8 entropy-24-01788-f008:**
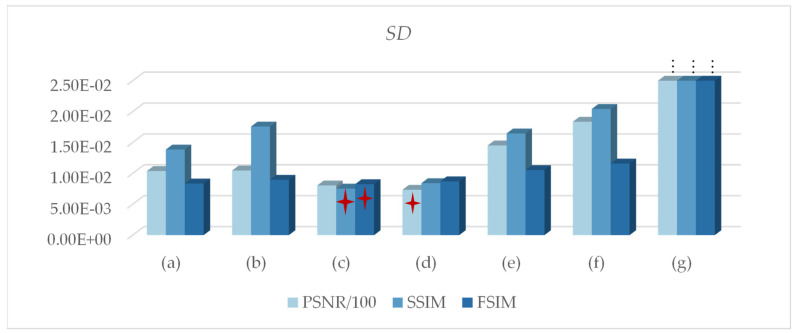
Standard deviation profile: (**a**) PERSSA-Renyi entropy; (**b**) SSA-Renyi entropy; (**c**) PERSSA-symmetric cross entropy; (**d**) SSA-symmetric cross entropy; (**e**) PERSSA-Kapur entropy; (**f**) SSA-Kapur entropy; (**g**) FCM. These three red stars are the optimal *SD* values on PSNR, SSIM, and FSIM respectively.

**Figure 9 entropy-24-01788-f009:**
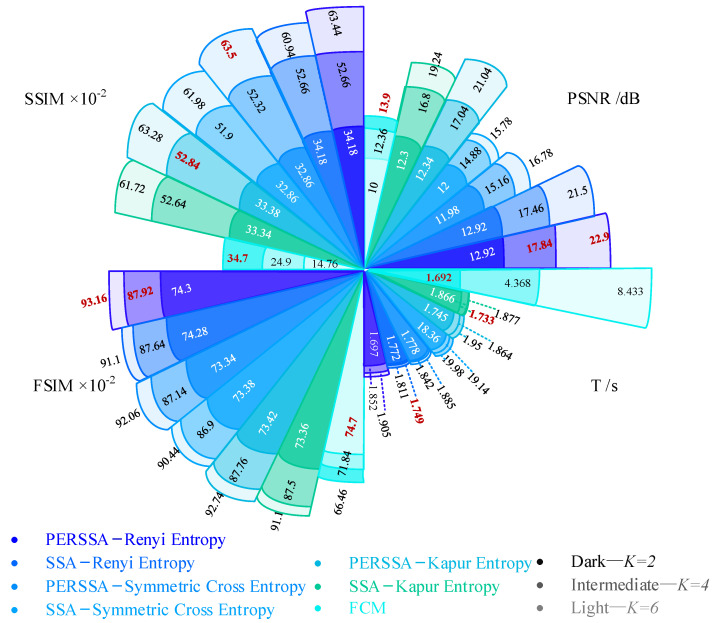
Fan charts of seven segmentation algorithms on four evaluation parameters. The red word is the optimal value under each *K* value and each parameter.

**Figure 10 entropy-24-01788-f010:**
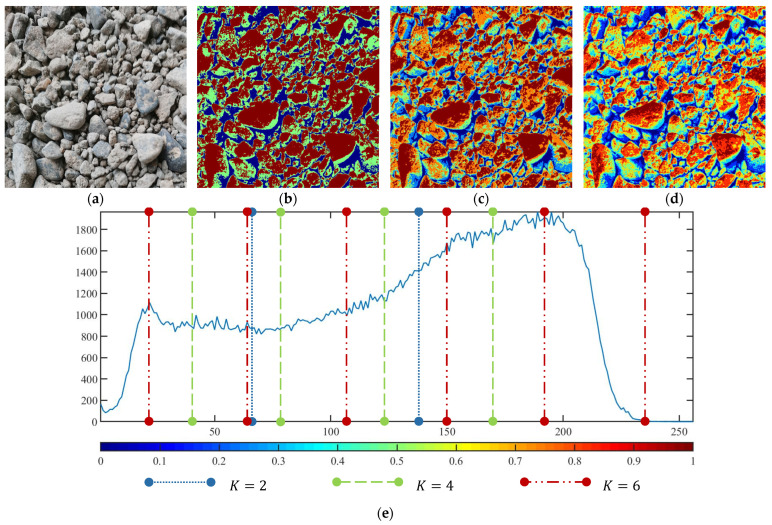
The segmentation result of PERSSA-Renyi entropy for No. 5: (**a**) original image; (**b**) color segmentation map with K=2; (**c**) color segmentation map with K=4; (**d**) color segmentation map with K=6; (**e**) segmentation threshold line.

**Table 1 entropy-24-01788-t001:** The segmentation results of aggregate images by three METs. The blue box is the area where the segmentation results are significantly different, and the red arrow is the extreme value in the histogram.

Original Image	Histogram	Renyi Entropy	Symmetric Cross Entropy	Kapur Entropy
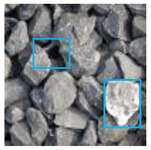	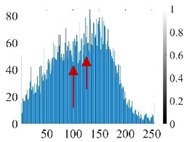	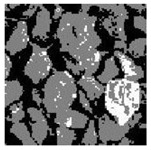	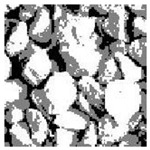	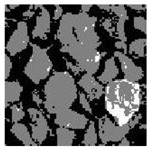
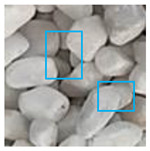	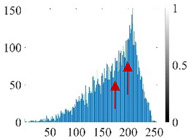	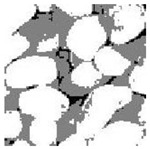	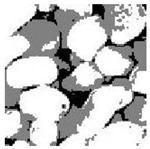	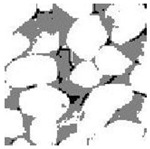

**Table 2 entropy-24-01788-t002:** Six benchmark functions.

Function	Name	Definition	Range	Optimum
Uni-modal benchmark functions	Sphere	$F_{1}\left(x\right)=\overset{D}{\underset{i=1}{\sum }}x_{i}^{2}$	${\left[-100,100\right]}^{D}$	0
	MaxMod	$F_{2}\left(x\right)=\max\left\{\left|x_{i}\right|,\;1\leq i\leq D\right\}$	${\left[-100,100\right]}^{D}$	0
	Rosen brock	$F_{3}\left(x\right)=\overset{D}{\underset{i=1}{\sum }}\left[100{\left(x_{i+1}-x_{i}^{2}\right)}^{2}+{\left(x_{i}-1\right)}^{2}\right]$	${\left[-30,30\right]}^{D}$	0
Multi-modal benchmark functions	Schwefel-26	$F_{4}\left(x\right)=\overset{D}{\underset{i=1}{\sum }}-x_{i}\cdot \sin\left(\sqrt{\left|x_{i}\right|}\right)$	${\left[-500,500\right]}^{D}$	−418.9829*D*
	Ackley	$F_{5}\left(x\right)=-20\exp\left(-0.2\sqrt{\frac{1}{D}\overset{D}{\underset{i=1}{\sum }}x_{i}^{2}}\right)-exp\left(\frac{1}{D}\overset{D}{\underset{i=1}{\sum }}\cos\left(2\pi x_{i}\right)\right)+20+e$	${\left[-32,32\right]}^{D}$	0
	Generalized Penalized	$F_{6}\left(x\right)=\frac{\pi }{D}\left\{10sin^{2}\left(\pi y_{1}\right)+\overset{D-1}{\underset{i=1}{\sum }}{\left(y_{i}-1\right)}^{2}\left(1+10sin^{2}\left(\pi y_{i+1}\right)\right)+\overset{D}{\underset{i=1}{\sum }}u\left(x_{i},\;10,\;100,\;4\right)\right\}$ $y_{i}=1+\left(x_{i}+1\right)/4$ $u\left(x_{i},\;a,\;k,m\right)=\{\begin{array}{cc} k{\left(x_{i}-a\right)}^{m} & x_{i}>a\\ 0 & -a\leq x_{i}\leq a\\ k{\left(-x_{i}-a\right)}^{m} & x_{i}<-a \end{array}$	${\left[-50,50\right]}^{D}$	0

**Table 3 entropy-24-01788-t003:** Parameter evaluations of the optimization algorithms on benchmark functions.

F ^1^	P ^2^	PSO	GWO	WOA	MA	SSA	CASSA	CDLSSA	PERSSA
Uni-modal benchmark functions									
$F_{1}$	*Best*	1.40E-10	4.73E-45	6.32E-127	1.82E-10	4.94E-299	**0**	**0**	**0**
	*Avg*	5.15E-09	7.10E-43	1.25E-109	1.12E-08	9.82E-40	7.39E-59	**0**	**0**
	*SD*	1.44E-08	1.40E-42	6.72E-109	3.06E-08	7.60E-48	4.79E-57	**0**	**0**
	*T*	7.00E+01	6.32E+01	2.29E+01	9.53E+01	1.98E+01	**1.82E+01**	4.06E+01	2.06E+01
$F_{2}$	*Best*	6.20E-01	5.56E-12	3.58E-04	3.32E+01	2.19E-14	3.84E-27	1.51E-259	**0**
	*Avg*	1.54E+00	8.40E-10	1.62E+01	5.25E+01	6.93E-10	4.59E-13	2.87E-200	**0**
	*SD*	7.85E-01	1.74E-10	1.65E+01	8.05E+00	4.11E-09	3.43E-12	**0**	**0**
	*T*	6.74E+01	5.98E+01	2.30E+01	8.17E+01	**1.99E+01**	2.04E+01	4.04E+01	2.04E+01
$F_{3}$	*Best*	1.68E+01	2.52E+01	2.62E+01	1.89E+00	2.71E-04	7.87E-05	7.04E-03	**1.16E-09**
	*Avg*	2.19E+02	2.66E+01	2.72E+01	4.78E+01	5.21E-02	3.84E-02	1.41E-03	**4.45E-04**
	*SD*	6.64E+02	7.68E-01	7.02E-01	4.07E+01	1.05E-02	8.05E-02	4.42E-03	**8.99E-04**
	*T*	7.18E+01	6.36E+01	2.53E+01	1.05E+02	2.20E+01	**2.19E+01**	4.33E+01	2.26E+01
Multi-modal benchmark functions									
$F_{4}$	*Best*	−9.33E+03	−7.71E+03	**−1.26E+04**	−1.15E+04	−1.24E+03	−1.06E+03	−1.91E+03	**−1.26E+04**
	*Avg*	−8.22E+03	−6.21E+03	−9.01E+03	−1.08E+04	−1.04E+03	−9.18E+03	−1.88E+03	**−1.23E+04**
	*SD*	4.84E+02	8.06E+02	1.25E+03	3.66E+02	2.02E+03	1.68E+03	1.01E+02	**9.48E+01**
	*T*	7.26E+01	6.49E+01	2.57E+01	1.17E+02	**2.22E+01**	2.26E+01	4.35E+01	2.29E+01
$F_{5}$	*Best*	2.18E-06	1.51E-14	**8.88E-16**	4.47E-07	8.57E-14	8.98E-14	**8.88E-16**	**8.88E-16**
	*Avg*	5.59E-01	2.53E-14	4.09E-15	5.84E-02	8.47E-16	1.18E-15	**8.88E-16**	**8.88E-16**
	*SD*	6.74E-01	4.39E-15	2.75E-15	6.79E-02	4.59E-16	9.90E-16	**0**	**0**
	*T*	7.10E+01	6.28E+01	2.39E+01	9.22E+01	2.09E+01	**2.07E+01**	4.10E+01	2.16E+01
$F_{6}$	*Best*	6.28E-11	2.00E-06	5.45E-04	6.00E-08	5.62E-12	3.33E-13	4.65E-10	**6.58E-14**
	*Avg*	1.32E-01	2.52E-02	2.79E-02	5.36E-02	4.94E-07	4.13E-07	2.84E-06	**3.70E-08**
	*SD*	2.75E-01	1.48E-02	6.97E-02	9.09E-02	2.06E-06	1.27E-06	**1.25E-06**	1.52E-06
	*T*	9.95E+01	9.37E+01	5.13E+01	1.54E+02	4.86E+01	4.87E+01	8.85E+01	**4.86E+01**

^1^ F is the function. ^2^ P is the parameter.

**Table 4 entropy-24-01788-t004:** The Wilcoxon rank-sum test between PERSSA and the other algorithms.

Parameter	PERSSA- PSO	PERSSA- GWO	PERSSA- WOA	PERSSA- MA	PERSSA- SSA	PERSSA- CASSA	PERSSA- CDLSSA
*Best*	**7.16E-04**	**1.71E-02**	**1.35E-02**	**1.22E-03**	1.87E-01	4.84E-01	6.27E-01
*Avg*	**3.11E-04**	**5.04E-03**	**3.21E-03**	**1.94E-03**	1.20E-01	1.41E-01	6.33E-01
*SD*	**1.58E-05**	**4.62E-03**	**4.11E-04**	**3.10E-04**	**4.72E-02**	**4.73E-02**	9.12E-01
*T*	**1.72E-07**	**5.50E-03**	6.73E-01	**3.06E-08**	8.13E-01	8.77E-01	**3.21E-04**

**Table 5 entropy-24-01788-t005:** Aggregate images and their partial segmentation results. The blue box is the area with better segmentation results, and the red box is the area with severe segmentation result distortion.

	No. 1	No. 2	No. 3	No. 4	No. 5
Original image	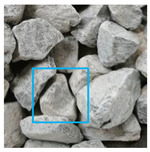	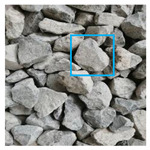	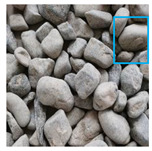	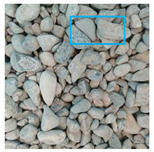	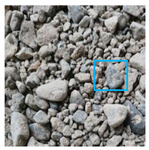
K	2	2	4	4	6
PERSSA-Renyi Entropy	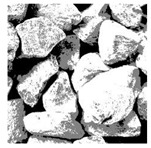	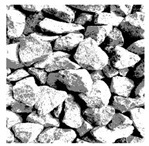	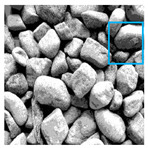	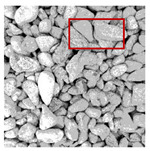	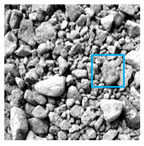
PERSSA-Symmetric Cross Entropy	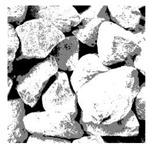	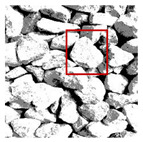	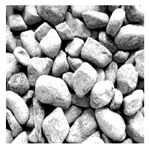	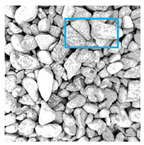	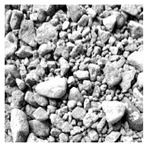
PERSSA-Kapur Entropy	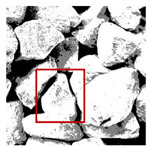	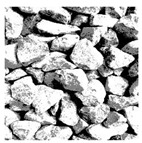	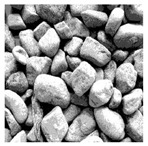	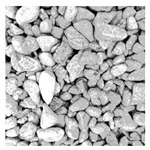	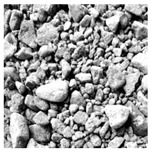
FCM	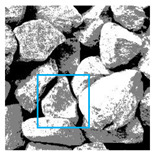	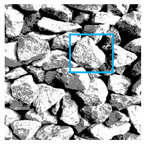	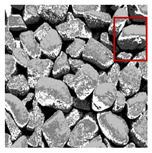	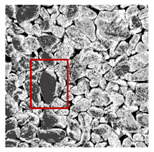	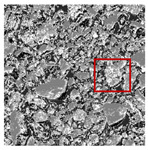

**Table 6 entropy-24-01788-t006:** Histograms of the aggregate image segmentation results.

	No. 1	No. 2	No. 3	No. 4	No. 5
Original image	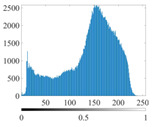	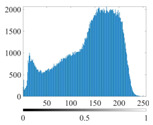	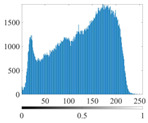	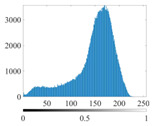	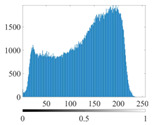
K	2	2	4	4	6
PERSSA-Renyi Entropy	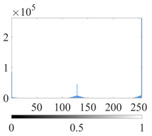	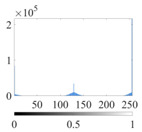	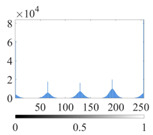	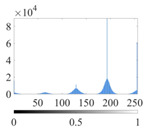	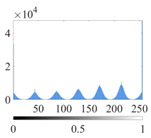
FCM	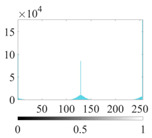	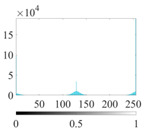	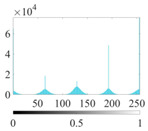	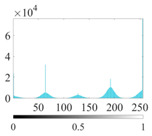	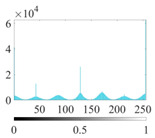

**Table 7 entropy-24-01788-t007:** Thresholds and fitness values searched by PERSSA and SSA on MET when K=6.

Value	Image	Renyi Entropy		Symmetric Cross Entropy		Kapur Entropy	
		PERSSA	SSA	PERSSA	SSA	PERSSA	SSA
Thresholds	No. 1	**41 78 114 146 173 205**	40 82 112 141 170 199	**27 53 87 123 157 189**	39 68 97 129 148 180	**38 70 102 134 169 198**	37 59 96 129 197 162
	No. 2	**44 79 116 154 198 231**	50 74 99 125 176 226	**26 48 82 114 152 186**	30 55 83 113 148 180	**41 91 127 162 189 233**	41 90 127 161 186 234
	No. 3	**41 80 109 153 186 229**	36 62 100 121 189 229	**32 65 94 122 148 181**	31 63 100 130 160 184	**46 86 122 155 188 230**	56 95 117 143 170 231
	No. 4	**39 76 115 151 188 223**	43 78 111 145 183 223	**31 58 94 125 156 182**	32 58 83 126 154 178	**49 86 125 156 190 227**	38 73 99 116 156 189
	No. 5	**36 67 101 136 168 198**	39 59 87 124 160 195	**29 56 90 119 152 187**	30 45 66 89 133 178	**37 69 98 127 158 191**	39 63 88 118 159 192
Fitness value	No. 1	**2.4450E+01**	2.4436E+01	**2.2693E+05**	2.4903E+05	**2.4245E+01**	2.4184E+01
	No. 2	**2.4799E+01**	2.4493E+01	**2.6417E+05**	2.6713E+05	**2.4424E+01**	2.4409E+01
	No. 3	**2.4708E+01**	2.4339E+01	**2.6368E+05**	2.6778E+05	**2.4437E+01**	2.4162E+01
	No. 4	**2.4325E+01**	2.4316E+01	**1.8345E+05**	1.9189E+05	**2.3906E+01**	2.3718E+01
	No. 5	**2.4371E+01**	2.4274E+01	**2.5295E+05**	2.9481E+05	**2.4156E+01**	2.4079E+01

**Table 8 entropy-24-01788-t008:** *PSNR* values.

Image	K	Parameter	Renyi Entropy		Symmetric Cross Entropy		Kapur Entropy		FCM
			PERSSA	SSA	PERSSA	SSA	PERSSA	SSA	
No. 1	2	*Avg*	1.27E+01	1.27E+01	1.19E+01	1.19E+01	1.13E+01	1.13E+01	1.44E+01
		*SD*	**4.36E-02**	1.59E-01	7.22E-02	**6.75E-02**	**2.42E+00**	3.10E+00	3.75E-03
	4	*Avg*	1.71E+01	1.71E+01	**1.50E+01**	1.49E+01	**1.67E+01**	1.66E+01	1.17E+01
		*SD*	**9.13E-01**	1.05E+00	1.82E-01	**1.23E-01**	2.21E+00	**1.24E+00**	2.26E+00
	6	*Avg*	**2.12E+01**	2.00E+01	**1.70E+01**	1.56E+01	**1.97E+01**	1.85E+01	1.08E+01
		*SD*	1.73E+00	**1.68E+00**	2.10E+00	**2.07E+00**	2.35E+00	**1.94E+00**	1.36E+01
No. 2	2	*Avg*	1.34E+01	1.34E+01	1.16E+01	1.16E+01	1.27E+01	1.27E+01	1.41E+01
		*SD*	9.24E-02	**6.21E-02**	**5.02E-02**	5.08E-02	**1.57E-01**	2.39E+00	2.32E-03
	4	*Avg*	**1.80E+01**	1.69E+01	1.50E+01	1.50E+01	**1.67E+01**	1.66E+01	1.38E+01
		*SD*	**1.10E+00**	1.17E+00	1.26E+00	**2.04E-01**	**1.55E+00**	1.56E+00	3.21E+00
	6	*Avg*	**2.36E+01**	2.12E+01	**1.60E+01**	1.53E+01	**2.18E+01**	2.17E+01	9.53E+00
		*SD*	**1.95E+00**	2.21E+00	2.23E+00	**1.63E+00**	**1.96E+00**	2.16E+00	3.62E+01
No. 3	2	*Avg*	1.35E+01	1.35E+01	**1.21E+01**	1.20E+01	**1.33E+01**	1.32E+01	1.34E+01
		*SD*	**9.13E-02**	1.67E-01	**5.18E-02**	5.40E-02	**2.21E-01**	2.27E+00	2.31E-03
	4	*Avg*	**1.78E+01**	1.72E+01	**1.50E+01**	1.44E+01	**1.73E+01**	1.71E+01	1.21E+01
		*SD*	1.30E+00	**1.21E+00**	2.17E-01	**1.96E-01**	1.33E+00	**1.11E+00**	5.23E+00
	6	*Avg*	**2.37E+01**	2.23E+01	1.62E+01	**1.76E+01**	**2.23E+01**	1.99E+01	1.04E+01
		*SD*	**1.95E+00**	2.22E+00	2.06E+00	**1.96E+00**	**2.24E+00**	2.59E+00	4.31E+01
No. 4	2	*Avg*	1.09E+01	1.09E+01	1.13E+01	**1.14E+01**	1.06E+01	1.06E+01	1.33E+01
		*SD*	**5.25E-02**	1.31E-01	5.48E-02	**4.20E-02**	3.57E-02	**3.37E-02**	1.36E-03
	4	*Avg*	**1.80E+01**	1.78E+01	**1.50E+01**	1.43E+01	**1.68E+01**	1.66E+01	1.05E+01
		*SD*	**8.56E-01**	8.88E-01	**1.92E-01**	2.87E-01	**7.46E-01**	1.63E+00	6.33E+00
	6	*Avg*	2.45E+01	2.45E+01	**1.67E+01**	1.59E+01	**2.23E+01**	1.73E+01	9.48E+00
		*SD*	1.87E+00	**1.58E+00**	**1.43E+00**	2.16E+00	**1.22E+00**	1.60E+00	5.26E+01
No. 5	2	*Avg*	1.41E+01	1.41E+01	1.30E+01	**1.31E+01**	**1.38E+01**	1.37E+01	1.43E+01
		*SD*	**4.32E-02**	4.33E-02	**1.01E-01**	1.16E-01	**8.31E-02**	9.78E-02	1.64E-03
	4	*Avg*	1.83E+01	1.83E+01	1.58E+01	1.58E+01	**1.77E+01**	1.71E+01	1.37E+01
		*SD*	**1.69E+00**	1.72E+00	**1.21E-01**	1.96E-01	**3.06E+00**	3.21E+00	4.36E+00
	6	*Avg*	**2.15E+01**	1.95E+01	**1.80E+01**	1.45E+01	**1.92E+01**	1.88E+01	9.80E+00
		*SD*	1.89E+00	**1.40E+00**	1.95E+00	**1.86E+00**	**2.19E+00**	2.61E+00	3.18E+01
Bold number			**17**	5	**14**	12	**23**	4	-
Optimal values number			**12**	6	2	**4**	**1**	0	9

**Table 9 entropy-24-01788-t009:** *SSIM* values.

Image	K	Parameter	Renyi Entropy		Symmetric Cross Entropy		Kapur Entropy		FCM
			PERSSA	SSA	PERSSA	SSA	PERSSA	SSA	
No. 1	2	*Avg*	3.00E-01	3.00E-01	2.91E-01	2.91E-01	2.83E-01	2.83E-01	3.12E-01
		*SD*	**2.03E-03**	2.52E-02	**4.66E-04**	5.11E-04	**3.13E-02**	3.64E-02	3.10E-04
	4	*Avg*	4.73E-01	**4.75E-01**	**4.69E-01**	4.65E-01	**4.72E-01**	4.71E-01	2.03E-01
		*SD*	**1.92E-02**	2.64E-02	**5.87E-03**	8.35E-03	**2.58E-02**	2.77E-02	6.25E-01
	6	*Avg*	**5.93E-01**	5.92E-01	**5.84E-01**	5.67E-01	**5.94E-01**	5.88E-01	1.25E-01
		*SD*	**1.39E-02**	1.82E-02	**1.54E-02**	1.72E-02	**2.10E-02**	2.25E-02	4.23E+00
No. 2	2	*Avg*	3.55E-01	3.55E-01	3.13E-01	3.13E-01	3.44E-01	3.44E-01	3.61E-01
		*SD*	1.54E-03	**8.58E-04**	**6.57E-04**	6.70E-04	**4.82E-03**	1.33E-02	2.98E-04
	4	*Avg*	**5.65E-01**	5.62E-01	**5.53E-01**	5.52E-01	**5.66E-01**	5.65E-01	2.52E-01
		*SD*	1.43E-02	**9.52E-03**	6.92E-03	**4.41E-03**	1.68E-02	**2.08E-02**	6.78E-01
	6	*Avg*	**6.65E-01**	6.20E-01	**6.69E-01**	6.57E-01	**6.62E-01**	6.58E-01	1.03E-01
		*SD*	**1.42E-02**	2.15E-02	2.65E-02	**1.85E-02**	1.97E-02	**1.51E-02**	3.28E+00
No. 3	2	*Avg*	3.18E-01	3.18E-01	**3.05E-01**	3.04E-01	**3.18E-01**	3.17E-01	3.17E-01
		*SD*	**1.78E-03**	3.87E-03	**4.31E-04**	4.72E-04	**7.19E-03**	2.10E-02	1.99E-04
	4	*Avg*	**5.01E-01**	4.98E-01	**4.91E-01**	4.80E-01	**5.02E-01**	5.00E-01	2.17E-01
		*SD*	**1.82E-02**	1.87E-02	**4.55E-03**	4.66E-03	2.45E-02	**1.58E-02**	5.23E-01
	6	*Avg*	**6.01E-01**	5.34E-01	6.02E-01	**6.11E-01**	**6.00E-01**	5.51E-01	1.44E-01
		*SD*	2.56E-02	**1.70E-02**	**1.62E-02**	2.03E-02	**1.66E-02**	2.95E-02	4.16E+00
No. 4	2	*Avg*	3.30E-01	3.30E-01	3.36E-01	3.36E-01	3.21E-01	3.21E-01	3.41E-01
		*SD*	**6.91E-03**	2.46E-02	**1.37E-03**	1.38E-03	**1.10E-03**	1.26E-03	3.32E-04
	4	*Avg*	4.97E-01	**5.01E-01**	**5.14E-01**	5.10E-01	5.05E-01	**5.08E-01**	2.80E-01
		*SD*	**2.02E-02**	2.14E-02	**4.10E-03**	6.35E-03	**1.97E-02**	2.45E-02	8.15E-01
	6	*Avg*	6.06E-01	**6.07E-01**	**6.21E-01**	6.16E-01	**6.07E-01**	5.97E-01	1.77E-01
		*SD*	2.55E-02	**1.91E-02**	**9.83E-03**	1.72E-02	2.19E-02	**2.17E-02**	2.59E+00
No. 5	2	*Avg*	4.06E-01	4.06E-01	3.98E-01	**3.99E-01**	**4.03E-01**	4.02E-01	4.04E-01
		*SD*	1.06E-03	**1.02E-03**	**3.51E-04**	3.65E-04	**2.42E-03**	3.11E-03	1.85E-04
	4	*Avg*	5.97E-01	5.97E-01	**5.89E-01**	5.88E-01	**5.97E-01**	5.88E-01	2.93E-01
		*SD*	**2.20E-02**	3.44E-02	7.33E-03	**6.52E-03**	1.87E-02	**1.74E-02**	6.21E-01
	6	*Avg*	**7.08E-01**	6.94E-01	**6.99E-01**	6.48E-01	**7.01E-01**	6.92E-01	1.89E-01
		*SD*	**2.14E-02**	2.19E-02	**1.30E-02**	1.93E-02	**1.53E-02**	3.63E-02	1.32E+00
Bold number			**16**	8	**22**	5	**21**	6	-
Optimal values number			**6**	4	**9**	3	**5**	0	**8**

**Table 10 entropy-24-01788-t010:** *FSIM* values.

Image	K	Parameter	Renyi Entropy		Symmetric Cross Entropy		Kapur Entropy		FCM
			PERSSA	SSA	PERSSA	SSA	PERSSA	SSA	
No. 1	2	*Avg*	7.00E-01	7.00E-01	6.95E-01	6.95E-01	6.85E-01	**6.86E-01**	7.07E-01
		*SD*	**1.27E-03**	1.35E-03	**1.27E-03**	1.48E-03	1.12E-02	**9.10E-03**	5.02E-04
	4	*Avg*	8.53E-01	8.53E-01	**8.46E-01**	8.43E-01	**8.53E-01**	8.52E-01	7.02E-01
		*SD*	**9.56E-03**	1.30E-02	**7.15E-03**	1.04E-02	**1.44E-02**	1.46E-02	6.14E-01
	6	*Avg*	**9.11E-01**	9.06E-01	**9.08E-01**	8.87E-01	9.13E-01	9.11E-01	6.68E-01
		*SD*	**1.04E-02**	1.09E-02	**1.81E-02**	1.95E-02	**1.14E-02**	1.24E-02	5.12E+00
No. 2	2	*Avg*	7.68E-01	7.68E-01	7.37E-01	7.37E-01	7.55E-01	7.55E-01	7.68E-01
		*SD*	7.79E-04	**1.17E-04**	8.22E-04	**7.59E-04**	**2.29E-03**	8.40E-03	5.68E-04
	4	*Avg*	**8.97E-01**	8.88E-01	**8.88E-01**	8.87E-01	**8.94E-01**	8.92E-01	7.57E-01
		*SD*	1.05E-02	**9.30E-03**	1.08E-02	**4.74E-03**	**1.30E-02**	1.31E-02	5.48E-01
	6	*Avg*	**9.44E-01**	9.04E-01	**9.29E-01**	9.18E-01	**9.32E-01**	9.28E-01	6.75E-01
		*SD*	**8.62E-03**	1.45E-02	2.49E-02	**1.82E-02**	1.26E-02	**1.19E-02**	3.25E+00
No. 3	2	*Avg*	**7.24E-01**	7.23E-01	**7.15E-01**	7.14E-01	**7.24E-01**	7.23E-01	7.23E-01
		*SD*	**4.27E-04**	1.48E-03	**1.21E-03**	1.30E-03	**3.45E-03**	1.00E-02	5.36E-04
	4	*Avg*	**8.67E-01**	8.62E-01	**8.59E-01**	8.53E-01	**8.67E-01**	8.63E-01	7.01E-01
		*SD*	1.29E-02	1.29E-02	**4.80E-03**	4.83E-03	1.49E-02	**1.25E-02**	7.35E-01
	6	*Avg*	**9.20E-01**	8.77E-01	9.04E-01	**9.07E-01**	**9.19E-01**	8.67E-01	6.88E-01
		*SD*	1.23E-02	**1.02E-02**	**1.76E-02**	1.92E-02	**1.27E-02**	1.67E-02	4.23E+00
No. 4	2	*Avg*	7.38E-01	7.38E-01	7.46E-01	**7.47E-01**	**7.28E-01**	7.26E-01	7.44E-01
		*SD*	**5.54E-04**	5.20E-03	**7.38E-04**	7.70E-04	**8.66E-03**	1.06E-02	4.63E-04
	4	*Avg*	**8.77E-01**	8.76E-01	**8.73E-01**	8.72E-01	8.75E-01	**8.76E-01**	6.90E-01
		*SD*	1.61E-02	**1.19E-02**	**2.96E-03**	4.09E-03	**1.08E-02**	1.27E-02	6.42E-01
	6	*Avg*	**9.37E-01**	9.32E-01	**9.23E-01**	9.14E-01	**9.34E-01**	9.15E-01	6.08E-01
		*SD*	**1.16E-02**	1.26E-02	**9.78E-03**	1.63E-02	1.61E-02	**1.42E-02**	3.56E+00
No. 5	2	*Avg*	7.85E-01	7.85E-01	7.74E-01	**7.76E-01**	**7.79E-01**	7.78E-01	7.93E-01
		*SD*	9.03E-04	**8.74E-04**	9.98E-04	**9.30E-04**	1.14E-03	**9.20E-04**	5.34E-04
	4	*Avg*	9.02E-01	**9.03E-01**	**8.91E-01**	8.90E-01	**8.99E-01**	8.92E-01	7.42E-01
		*SD*	**1.53E-02**	1.89E-02	**9.04E-03**	9.13E-03	1.22E-02	**1.06E-02**	5.13E-01
	6	*Avg*	**9.46E-01**	9.36E-01	**9.39E-01**	8.96E-01	**9.39E-01**	9.34E-01	6.84E-01
		*SD*	1.42E-02	**1.12E-02**	**1.35E-02**	1.90E-02	**1.31E-02**	1.62E-02	2.30E+00
Bold number			**17**	7	**21**	7	**20**	8	-
Optimal values number			**13**	6	**5**	2	**2**	0	5

**Table 11 entropy-24-01788-t011:** *T* values.

K	Renyi Entropy		Symmetric Cross Entropy		Kapur Entropy		FCM
	PERSSA	SSA	PERSSA	SSA	PERSSA	SSA	
2	**1.70**	1.77	**1.78**	1.84	**1.75**	1.87	1.69
4	1.85	**1.81**	**1.84**	2.00	1.95	**1.73**	4.37
6	1.91	**1.75**	**1.89**	1.91	**1.86**	1.88	8.43

**Table 12 entropy-24-01788-t012:** OMWP values.

Algorithm	PERSSA	SSA
Renyi Entropy	**8.79E-01**	8.33E-01
Symmetric Cross Entropy	8.05E-01	7.79E-01
Kapur Entropy	7.96E-01	7.23E-01
FCM	3.96E-01	

## Data Availability

The datasets analyzed during the current study are available in the CEC-Benchmark-Functions repository (https://github.com/tsingke/CEC-Benchmark-Functions), accessed on 17 July 2020.

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
