# Peer review of "Optimizing Multiple Entropy Thresholding by the Chaotic Combination Strategy Sparrow Search Algorithm for Aggregate Image Segmentation"

_entropy, 2022, doi:10.3390/e24121788_

Round 1
Reviewer 1 Report
Please see the attachment.

Author Response
请参阅附件。

Reviewer 2 Report
This is an article with a lot of effort put into it. However, the expression is very vague and there are many questions about the effective visual performance and indicators for comparing the results.
1. explanation addition to should be enhanced.
Line 53-54
Line 65-67
2. In eq. (1)~(6), make sure not to omit the parameter descriptions, R, S, K ? in subscripts
3. In 2.2 SSA, express the optimization result so that readers can see it as a picture.
4. In eq. (1), a non-generalized expression for the range of P. need to be corrected.
5. In fig.1, indicate the axis information.
6. In eq. (13), if the values of x_best and x_worst are not inversion, please delete the absolute value notation to avoid ambiguity.
7. In fig.2, correct the number to eq.(n).
8. In fig.4, Y-axis values are very strange. correct it.
9. In tables, e-(n), Expressions are different. Please note that it is unified. The E-01 notation seems unnecessary.
10. Correct Tbale in some tables.
11. In table 5, by what criteria can you judge that segmentation is well accomplished in the images? A clearer explanation is needed for colored boxes. (Major point)
12. In table 6, correct the figure values(totally expressions) to make them more understandable.
13. In table 7, as table 5, the definition and improvement criteria for particle segmentations should be clear. What criteria do you visually judge? (Major point)
14. Please rewrite Tables 5, 6, and 7 clearly to be understand.
15. Does PERSSA in the performance label mean PERSSA-MET? Please fix it.
16. In table 12, FCM values are exactly same. Right?
Reviewer 3 Report
1. Why choose Sparrow Search Algorithm for optimization?
2. How to apply Sparrow Search Algorithm to optimize multiple entropy thresholding is not clear.
Round 2
Reviewer 1 Report
Please double check the typos and minor mistakes. Also sentence structure.
Reviewer 2 Report
Acceptable in present form
Reviewer 3 Report
This research proposes an autonomous segmentation model 15 (PERSSA-MET) that optimizes MET based on the chaotic combination strategy Sparrow Search Algorithm (SSA). English is difficult to understand. Extensive editing of English language and style required.